# Microarchitectural Study of the Augmented Bone Following a Modified Ridge Splitting Technique: Histological and Micro-Computed Tomography Analyses

**DOI:** 10.3390/jcm13226679

**Published:** 2024-11-07

**Authors:** Dorottya Pénzes, Csilla Szerencse, Martin Major, György Szabó, Endre Kontsek, János Báskay, Péter Pollner, Bence Tamás Szabó, Csaba Dobó-Nagy, Dániel Csete, Attila Mócsai, Nadim Z. Baba, Orsolya Németh, Márton Kivovics, Eitan Mijiritsky

**Affiliations:** 1Department of Public Dental Health, Semmelweis University, Szentkirályi utca 40, 1088 Budapest, Hungary; penzes.dorottya@semmelweis.hu (D.P.); szerencse.csilla@semmelweis.hu (C.S.); nemeth.orsolya@semmelweis.hu (O.N.); 2Department of Oro-Maxillofacial Surgery and Stomatology, Semmelweis University, Mária utca 52, 1085 Budapest, Hungary; major.martin@phd.semmelweis.hu (M.M.); szabo.gyorgy2@semmelweis.hu (G.S.); 3Department of Pathology, Forensic and Insurance Medicine University, Üllői út 93, 1091 Budapest, Hungary; 4Data-Driven Health Division of National Laboratory for Health Security, Health Services Management Training Centre, Semmelweis University, Kútvölgyi út 2, 1125 Budapest, Hungary; 5Department of Biological Physics, Eötvös Loránd University, Pázmány Péter Sétány 1/a, 1117 Budapest, Hungary; 6Department of Oral Diagnostics, Semmelweis University, Szentkirályi utca 47, 1088 Budapest, Hungary; szabo.bence.tamas@semmelweis.hu (B.T.S.); dobo-nagy.csaba@semmelweis.hu (C.D.-N.); 7Department of Physiology, Semmelweis University, Tűzoltó u. 34-37, 1094 Budapest, Hungary; csete.daniel@semmelweis.hu (D.C.); mocsai.attila@semmelweis.hu (A.M.); 8Advanced Dental Education Program in Implant Dentistry, School of Dentistry, Loma Linda University, Loma Linda, CA 92350, USA; nbaba@llu.edu; 9Department of Head and Neck Surgery and Maxillofacial Surgery, Tel-Aviv Sourasky Medical Center, School of Medicine, Tel Aviv University, Tel Aviv 64239, Israel; mijiritsky@bezeqint.net; 10Goldschleger School of Dental Medicine, Faculty of Medicine, Tel Aviv University, Tel Aviv 39040, Israel

**Keywords:** alveolar ridge splitting, alveolar ridge expansion, alveolar ridge augmentation, autologous bone, micro-CT, histomorphometry

## Abstract

**Objectives**: The aim of this matched prospective cohort study was to examine the microarchitecture of the augmented bone following a modified alveolar ridge splitting procedure and compare it to that of native bone. **Methods**: In the test group, patients underwent a modified ridge split osteotomy procedure to restore the width of the posterior segment of the mandible. Patients with sufficient bone width for dental implant placement in the posterior region of the mandible following 3-month-long spontaneous healing after tooth removal were included in the control group. In both study groups, bone biopsy samples were harvested and dental implants were placed. Histomorphometry and micro-CT analysis were performed. **Results**: Altogether, 15 patients were included in this study (7 patients in the test group, with 14 bone core biopsies harvested, and 8 patients in the control group, with 13 bone core biopsies harvested). Percentage bone volume (BV/TV) in the micro-CT analysis (22.088 ± 8.094% and 12.075 ± 4.009% for the test and control group, respectively) showed statistically significant differences between study groups. **Conclusions**: Based on histological and micro-CT analyses, the modified ridge splitting procedure with autologous bone block harvested from the retromolar area results in a dental implant recipient bone microarchitecture superior to that of the extraction sockets left to heal undisturbed for a 3-month-long healing period.

## 1. Introduction

The implant-borne fixed partial denture is a safe and predictable option to rehabilitate the edentulous premolar and molar region of the mandible [1,2,3]. Following tooth removal, immediate implant placement may preserve the alveolar anatomy [4]. However, in the case of late implant placement, alveolar atrophy may result in a horizontally deficient ridge [5]. In these cases, lateral augmentation of the edentulous ridge prior to or at the time of implant placement is necessary for the survival and long-term stability of the implant-borne restoration [6].

Guided bone regeneration and onlay block grafts have been applied successfully to restore the horizontally deficient alveolar ridge [7,8,9,10,11]. Tatum was the first to introduce the alveolar ridge splitting procedure as a means of alveolar ridge expansion. Alveolar ridge splitting is a safe and predictable method to augment the horizontally atrophied alveolar ridge [12]. This technique may be utilized to increase the width of bone by splitting the vestibular cortical plates and expanding the narrow residual ridge [13]. Alveolar ridge splitting may be performed at the time of implant placement with or without the use of bone graft materials and membranes, or prior to implant placement [14,15,16].

In a previous study, our research group reported on the clinical results of a novel, modified ridge splitting procedure using autologous bone blocks from the retromolar area [12]. This modified approach enables the clinician to access the recipient and donor site from a single flap, eliminating the need for any other biomaterials except for the autologous bone block and the osteosynthesis screws used to stabilize the blocks. The novel method decreases the costs of the intervention and shortens the healing period to 3 months before dental implant placement. However, increased morbidity may be expected due to the autologous bone harvested from the retromolar area. Another limitation of the modified technique may be that it is staged; dental implant placement is performed at a later appointment following bone healing [12].

At the time of dental implant placement, not only recipient bone quantity but also the quality is paramount for implant survival [17,18,19]. Therefore, it is important to assess the microarchitecture of the augmented bone and compare it to the native bone when reporting on a modified bone augmentation technique.

Currently, histomorphometry is the gold standard for bone microarchitectural analysis [20,21]. Histomorphometry provides information on cellularity and tissues. However, it is a destructive process and does not allow for the assessment of the three-dimensional trabecular structure of the bone. Micro-computed tomography (micro-CT) allows direct three-dimensional analysis of bone structures; therefore, it is considered in the literature as an additional non-destructive imaging modality to study trabecular microarchitecture [22].

The aim of this matched prospective cohort study was to evaluate the microarchitecture of the augmented bone following the modified alveolar ridge splitting procedure and compare it to that of native bone using histomorphometry and micro-CT analysis.

## 2. Materials and Methods

### 2.1. Study Design

This matched prospective cohort study was approved by the Medical Research Council Committee of Science and Research Ethics (ETT TUKEB 5308-7/2019/EÜIG) and the Office of the Chief Medical Officer of The National Public Health and Medical Officer Service (IF-14561-10/2015) (18 December 2015). The study was registered at ClinicalTrials.gov (NCT05005858), and it was performed in adherence with the Helsinki Declaration [23]. Patients were informed about the surgical interventions and analyses performed as part of this study and signed the necessary consent forms before treatment. This study was designed following STROBE guidelines [24,25].

Periodontally healthy patients of Semmelweis University’s Department of Community Dentistry who were over 18 years old and had Kennedy Class I and II mandibles were included in our study. Patients with inadequate bone width for implant placement at the premolar and molar region of the mandible were included in the test group.

Anatomical inclusion criteria for the test group were as follows:Mandibular bone width of at least 3 mm.Bone height of at least 11 mm.Cancellous bone present between the buccal and lingual cortical plates at least 1 mm in width [26].Healing period of at least 3 months following extractions in the premolar and molar regions of the mandible.

In the test group, patients underwent a modified ridge split osteotomy procedure to restore the width of the posterior segment of the mandible.

Patients with sufficient horizontal dimensions for implant placement in the posterior mandible following 3-month-long spontaneous healing following tooth removal were included in the control group. Unrestorable teeth, teeth with hopeless prognosis due to periodontal disease, endodontic failure, and vertical root fracture were the indications of tooth removal. Patients in the control group were included from a pool to match the age distribution of the patients included in the test group. 

Anatomical inclusion criteria for the control group were as follows:Mandibular bone width of at least 7 mm.Bone height of at least 11 mm.Healing period of 3 months following extractions in the premolar and molar regions of the mandible at implant recipient sites.

Exclusion criteria for the test and control group were as follows:Psychiatric conditions that contraindicate rehabilitation with implant-borne prostheses.Uncontrolled medical disorders that contraindicate elective surgical interventions (hematological disorders or diseases, etc.).History of systemic diseases or medication that alter bone metabolism (alcoholism, dialysis, history of chronic hepatitis or liver cirrhosis, uncontrolled diabetes mellitus, etc.).History of medication that may alter bone metabolism (steroids, antiresorptive drugs (bisphosphonates, RANKL inhibitor antibodies), vascular endothelial growth factor (VEGF) antagonists, etc.).Immunosuppressive therapy.Chemotherapy.Radiotherapy of the head and neck region.Local inflammations, cysts, and tumors at the site of the planned surgical intervention.Non-compliance and inability to attend follow-ups.Poor oral hygiene.Pregnancy.Smoking.

The recruitment, administration of surgical interventions, and evaluation of the outcome measures were continuous for this study. Sample size calculation was performed using the G*Power 3.1 software (v.3.1.9.3, 2017, Institut für Experimentelle Psychologie, Heinrich-Heine-Universität, Düsseldorf, Germany); 0.1 (10%) difference in BV/TV was considered as clinically significant. If power was set at 0.8 and α at 0.05, a sample size of 9 cases (bone core biopsy samples) per study group was calculated as sufficient.

### 2.2. Surgical Interventions

All surgical interventions were performed by the same surgeon, experienced in dental implant placement and bone management (D.P.).

#### 2.2.1. Alveolar Ridge Splitting in the Test Group

Before surgery, patients rinsed with a 0.2% chlorhexidine solution for 1 min. Under local anesthesia, following full-thickness flap preparation, a midcrestal osteotomy was performed. Two vertical releasing osteotomies were performed at the mesial and distal ends of the midcrestal osteotomy. Apically, a superficial corticotomy was carried out to connect the vertical osteotomies horizontally. Bone management was performed using a piezoelectronic device (NSK Variosurg3 Ultrasonic Bone Surgery System, NSK Europe GmbH, Eschborn, Germany). These osteotomies enabled the green-stick fracture of the buccal cortical. With this mobilization of the buccal cortical, we developed the recipient site for the bone block between the buccal and lingual cortical plates (Figure 1B). The autologous bone was harvested from the retromolar area (Figure 1A) from the same flap used to access the recipient site. The bone block was immobilized at the recipient site using osteosynthesis screws (MSS10 100; Meisinger Screw System, Hager and Meisinger GmbH, Neuss, Germany) (Figure 1C). Tension-free primary closure was obtained by mobilizing the lingual and buccal flaps. Two-layer flap closure was performed with horizontal mattress sutures and single interrupted sutures. Suture removal took place after 14 days [12]. 

Antibiotics (amoxicillin and clavulanate), two times a day for 5 days (Aktil Duo 875 mg/125 mg, Sandoz Hungária Kft., Budapest, Hungary) (in case of amoxicillin allergy, clindamycin 4 times a day for 4 days (Dalacin 300 mg, Pfizer Inc., New York, NY, USA)), a non-steroid anti-inflammatory drug, diclofenac, 3 times a day for 3 days (Cataflam 50 mg, Novartis Hungária Kft., Budapest, Hungary), and 0.2% chlorhexidine mouth rinse, two times a day (Corsodyl, GlaxoSmithKline Consumer Healthcare GmbH & Co. KG, München, Germany), were prescribed to the patients. During the 3-month-long healing period, patients did not wear temporary prostheses [12]. Figure 2 presents one of the cases in the test group.

#### 2.2.2. Tooth Extraction in the Control Group

All patients rinsed with a 0.2% chlorhexidine solution for 1 min before tooth removal. Under local anesthesia, the gingival attachment around the tooth was elevated with dental elevator. The tooth was removed using luxators and forceps in an atraumatic manner. Multi-rooted teeth were sectioned before removal. After the tooth removal, debridement was performed with a surgical curette, and the wound was rinsed using sterile saline solution. The site was closed with single interrupted sutures. Suture removal took place after 7 days. A non-steroid anti-inflammatory drug, diclofenac, 3 times a day for 3 days (Cataflam 50 mg, Novartis Hungária Kft., Budapest, Hungary), was prescribed to the patients. During the 3-month-long healing period, patients did not wear temporary prostheses. 

#### 2.2.3. Bone Core Biopsy Samples Harvesting

In both study groups, dental implants were inserted under local anesthesia from a full-thickness flap. Bone core biopsy samples, 8 mm in length, were harvested from the implant recipient sites, using a trephine drill (external diameter of 3.0 mm and an internal diameter of 2.0 mm (330 205 486 001 020 Hager and Meisinger GmbH, Neuss, Germany)) at a drill rotation speed of 800 rpm (Figure 3) [12].

Following bone biopsy, implant beds were prepared in adherence to the instructions of the implant manufacturer. Submerged dental implant placement (Nobel Replace Conical Connection, Nobel Biocare AG, Kloten, Switzerland) was performed (Figure 4). Implant uncovering and healing abutment connection were carried out 3 months after implant placement.

The bone biopsy samples were submerged in 10% formaldehyde solution in 0.1 M phosphate-buffered saline (PBS), pH 7.3, stored at 4 °C. Containers were marked with a unique identification number to enable blind histomorphometric and micromorphometric analyses.

### 2.3. Micro-CT Imaging and Micromorphometric Analysis

Bone biopsy specimens were scanned using a micro-CT scanner (Bruker 1272 X-ray microtomograph, Bruker μCT, Kontich, Belgium). Scanning was performed at a resolution of 11.0 μm (60 kV, 66 μA). A 0.25 mm aluminum filter was used for image noise reduction. The average scan duration was 20 min. The reconstruction of raw images was performed using NRecon software (v.1.7.4.6., Bruker μCT, Kontich, Belgium) with a ring artifact correction value of 2 and beam-hardening correction value of 0%. Quantitative micromorphometric analysis of the bone core biopsy specimens of the test and control groups was carried out with the CTAn software (v.1.17.7.2, Bruker μCT, Kontich, Belgium). The following 3D morphometric variables were selected and calculated: bone volume (BV), tissue volume (TV), and volume of interest (VOI) detected for the micro-CT measurement. Percent bone volume (BV/TV), bone surface/volume ratio (BS/TV), trabecular thickness (Tb.Th.), trabecular separation (Tb.Sp.), trabecular bone pattern factor (Tb.Pf), structure model index (SMI), total porosity (Po(tot)), and connectivity (Conn.) were the morphometric variables determined in our study. The micromorphometric data variables and their definitions are listed in Table 1. 

### 2.4. Histological Processing and Scanning

The bone core biopsy was embedded in paraffin (FFPE) after a 4-day-long decalcification. The biopsy was laid between tissue foam pads to obtain the horizontal embedding orientation. Sections of 5 µm thickness were obtained and placed on glass slides. The samples were routinely stained with hematoxylin and eosin (HE). Scanning was performed using a 3DHistech Pannoramic^®^ 1000 Digital Slide Scanner (3D Histech Kft., Budapest, Hungary). 

An in-house artificial intelligence (AI) application was used to classify tissues (trabecular bone and bone marrow) in the histological sections [28]. Three representative sections from each bone biopsy sample were evaluated, and the areas of trabecular bone and bone marrow were measured using the AI model. An expert (E.K.) supervised the automatic segmentation and made sure that the tissues were identified correctly. Bone volume percent (BVP) was determined as the primary outcome measure of the histomorphometric analysis which is the ratio (%) of the area of bone within the total area of the histological section. BVP was calculated as the mean of the measurements performed on the three representative histological sections for each bone biopsy specimen. 

### 2.5. Statistical Analysis

SPSS 28.0.1.0 software for Windows (SPSS, Inc., Chicago, IL, USA) was used for statistical analysis of micromorphometric and histomorphometric data. 

To test the normality of the distribution of micromorphometric and histomorphometric data, Shapiro–Wilk tests, visual inspection of histograms, and normal Q-Q plotting were carried out. Levene’s test was carried out to test the equal variance of the data. In the case of approximately normal distribution of the data, the independent sample t-test was performed. The Mann–Whitney test was used as a non-parametric test for the comparison of recorded measurements. 

Values of *p* < 0.05 were considered statistically significant.

The clinician responsible for the surgical interventions (D.P.) was not involved in the measurement of histomorphometric and micromorphometric outcome measures. The experts conducting the histomorphometric (E.K., J.B., P.P.) and micromorphometric (B.T.SZ., CS.D-N.) measurements were blinded to the surgical interventions performed. Statistical analysis of the data was performed by a blinded investigator (M.K.).

## 3. Results

A total of fifteen patients (eight female, seven male) seeking dental implant-supported prosthodontic rehabilitation were included in this study. None of the patients enrolled had diabetes mellitus in their medical history.

The flow diagram for this study is presented in Figure 5. Seven patients were included in the test group (two males, five females; mean age of 54.86 years ± 9.62 years; 14 bone core biopsy samples harvested), while eight patients were allocated to the control group (five males, three females; mean age 53.25 years ± 15.06 years; 13 bone core biopsy samples harvested). Demographic data of the patients included in this study are presented in Table 2.

During the 48 ± 12-month-long follow-up, one implant failed in the test group after 6 months and no implant failure was observed in the control group. The raw data for this study are presented in Appendix A.

### 3.1. Qualitative Results of the Micro-CT and Histological Analysis

The micro-CT images and histological sections of the bone core biopsy samples (Figure 6) showed that in the test group, the cortical bone block was remodeled into trabecular bone at the recipient site in all cases. Samples of the control group consisted of trabecular bone without any signs of inflammation (Figure 7). 

### 3.2. Quantitative Results of the Micro-CT Analysis 

The morphometric parameters of the augmented bone in the study group and the native bone of the control group were compared (Table 3). BV/TV, SMI, Po(tot), and Conn. showed statistically significantly different values between the test and control group. BV/TV (22.088 ± 8.094% and 12.075 ± 4.009% for test and control group, respectively) and Conn. values (2451.786 ± 1524.472 and 1040.154 ± 708.990 for test and control group, respectively) of the test group were significantly higher than those of the control group. Po(tot) (77.912 ± 8.094% and 87.925 ± 4.009% for test and control group, respectively), Tb.Sp (664.467 ± 218.084 µm and 990.254 ± 366.158 µm for test and control groups, respectively), and SMI (0.477 ± 1.826 and 1.954 ± 1.141 for test and control group, respectively) values of the test group were significantly lower than those of the control group. 

### 3.3. Histology and Histomorphometry

Sections of the test group (47.64 ± 10.77%) showed a tendency for higher BVP values than those of the control group (42.70 ± 10.72%); however, these differences were not statistically significant. Table 4 presents the results of the histomorphometric analysis.

The qualitative histological and micro-CT analyses performed in this study showed that ridge splitting using the surgical method described in this study leads to the remodeling of the autologous bone block into trabecular bone at the recipient site. According to the histomorphometric results, BVP of the augmented bone (47.64 ± 10.77%) following a 3-month healing period was higher than that of the extraction sockets (42.70 ± 10.72%) following the same healing period, which indicates higher bone density. However, unlike BV/TV data of the quantitative micro-CT analysis, this difference did not reach statistical significance. Higher bone density achieved by the modified ridge splitting method following a 3-month-long healing time was supported by the micromorphometric results, which indicate that BV/TV values of the augmented bone (22.088 ± 8.094%) are significantly higher than those of the healing extraction sockets (12.075 ± 4.009%).

## 4. Discussion

The aim of bone augmenting strategies is to regenerate bone which has a similar quality to native bone. In this study, native bone formed after a 3-month-long healing time following extraction was used as a control to match the healing time of the ridge splitting procedure because in the molar region, type 3 implant placement is a viable and frequently used timing of implant surgery [29]. 

During histomorphometric assessment, the pathologist measures the area occupied by different tissues on a representative section of the bone core biopsy sample. However, histomorphometry alone is not reliable in the evaluation of three-dimensional trabecular morphology [30,31]. Moreover, histomorphometric results are influenced by the thickness and orientation of the histologic sections [32,33,34]. Contrary to histomorphometry, micro-CT analysis is a reliable method for the direct three-dimensional evaluation of trabecular microarchitecture [34,35,36]. The higher bone density of the augmented sites compared to the extraction sockets is confirmed by the significantly higher Po(tot) (77.912 ± 8.094% and 87.925 ± 4.009% for test and control groups) and Tb.Sp (664.467 ± 218.084 µm and 990.254 ± 366.158 µm for test and control groups) values of the control group. Significantly higher Conn. values in the test group may be explained by the more interconnected trabecular structure of the augmented bone following ridge splitting osteotomy due to the nature of bone remodeling. Healing of the autologous bone block at the recipient site is a resorptive process overall, creating cancellous bone from cortical bone. Conversely, healing of the extraction socket results in new bone formation from the blood clot into the cancellous bone of the mandible. SMI values indicate that following ridge splitting, a more plate-like anatomy can be observed because of the remodeling pattern of the cortical bone block, whereas bone formation in the healing sockets results in a rod-like trabecular structure originating from the cancellous bone of the alveolar ridge.

According to the literature, insufficient bone quality and lower bone density at the implant recipient site are risk factors of early implant failure [17,18,19]. Therefore, improved bone quality at the augmented site prior to implant placement may prevent early implant failure and benefit patients. 

Bone remodeling following any kind of surgery is a long physiological process that lasts 4–6 months [37]. Further studies may be necessary to determine how bone microarchitecture changes during this time frame following ridge splitting and tooth removal. However, according to the results of the present study, the 3-month-long healing period following ridge splitting prior to dental implant placement determined for this study results in an ideal microarchitecture of recipient bone.

Several similar studies were described in the literature. Moro et al. presented alveolar ridge splitting using only autologous bone block harvested from the mandible [38]. Holtzclaw et al. [39], Blus et al. [40], and Basa et al. [41] described a similar alveolar ridge splitting technique. The success of these studies was determined by alveolar ridge width gain. Scipioni et al. [42] observed new bone formation following a graftless edentulous ridge expansion based on the results of histologic analysis of bone core biopsy samples. To the best of our knowledge, the present study is the first to evaluate the quality and microarchitecture of augmented bone following staged alveolar ridge splitting using autologous bone blocks, based on histomorphometric and micro-CT analyses. 

The few number of cases included may have been a limitation of this study. Another limitation of this study was that parameters of trabecular bone microarchitecture were not correlated with the sex and age of the patients enrolled because of the small sample size. A further limitation was that patient-reported postoperative pain was not recorded following surgical interventions. The strength of this study was its matched prospective cohort design which enabled the comparison of the microarchitecture of the augmented bone to that of the healing extraction socket, as type 3 implant placement is a safe and predictable treatment timing for the rehabilitation of posterior regions of the mandible with a low rate of implant failure [37].

## 5. Conclusions

Based on histological and micro-CT analyses, the modified ridge splitting procedure with autologous bone block harvested from the retromolar area of the mandible results in a dental implant recipient bone microarchitecture superior to that of the extraction sockets left to heal undisturbed after a 3-month-long healing period.

## Figures and Tables

**Figure 1 jcm-13-06679-f001:**
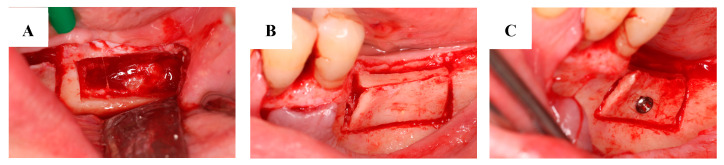
Harvesting a bone block from the retromolar area. The donor site after removing the autologous bone block (**A**). The autologous bone block placed between the lingual and cortical plates at the recipient site (**B**). The fixation of the bone block (**C**).

**Figure 2 jcm-13-06679-f002:**
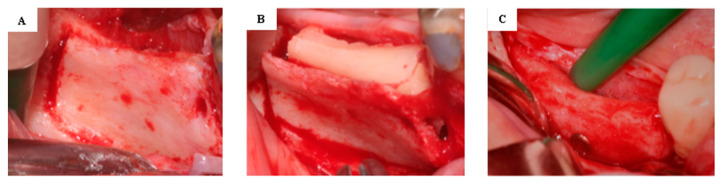
The modified ridge splitting procedure. Osteotomies and corticotomies performed prior to ridge splitting (**A**). The autologous bone block placed between the lingual and cortical plates (**B**). The alveolar ridge after the 3-month healing time (**C**).

**Figure 3 jcm-13-06679-f003:**
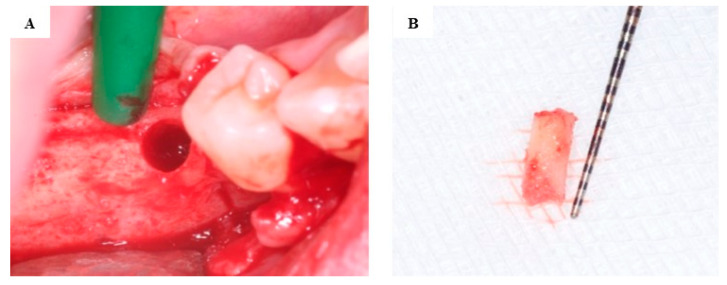
Bone core biopsy. Implant site following the use of the trephine drill (**A**). The bone core biopsy sample harvested during dental implant placement (**B**).

**Figure 4 jcm-13-06679-f004:**
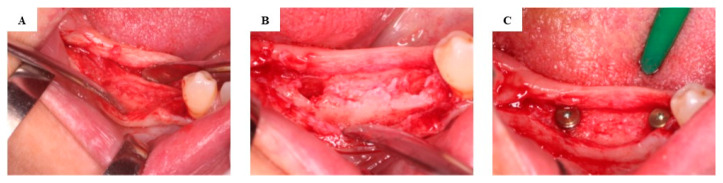
Re-entry in the test group after 3 months of healing. The augmented alveolar ridge following the elevation of the full-thickness flap (**A**). Implant osteotomies (**B**). The dental implants placed in the augmented bone (**C**).

**Figure 5 jcm-13-06679-f005:**
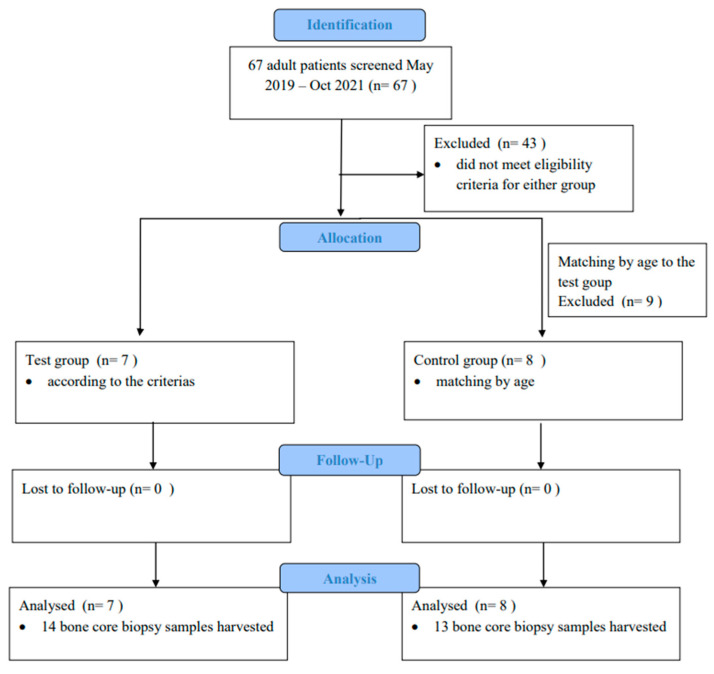
Flow diagram of the study.

**Figure 6 jcm-13-06679-f006:**
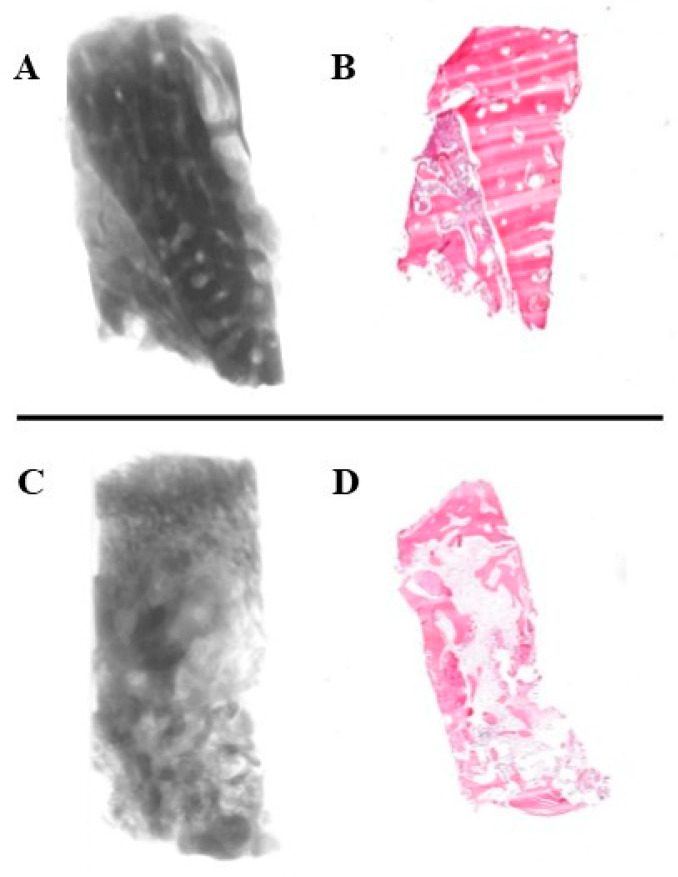
Micro-CT and histological appearance of the bone biopsies. Micro-CT reconstruction (**A**) and histological section (**B**) of a bone core biopsy sample harvested from the augmented area following alveolar ridge splitting (test group). Micro-CT reconstruction (**C**) and histological section (**D**) of a bone core biopsy sample harvested from a healed extraction socket (control group).

**Figure 7 jcm-13-06679-f007:**
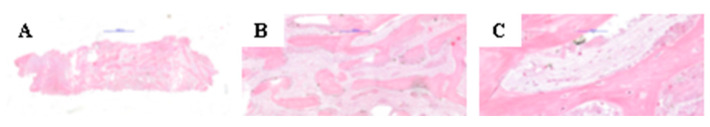
Histological appearance of the augmented bone of the control group without any signs of inflammation. Magnification 2× (**A**), 10× (**B**), and 40× (**C**).

**Table 1 jcm-13-06679-t001:** The micromorphometric variables and their abbreviations, definitions, and measurement units assessed in this study [27].

Variable	Abbreviation	Description	Unit
Percent bone volume	BV/TV	Relative volume of calcified tissue in the region of interest.	%
Bone surface/volume ratio	BS/TV	Segmented bone surface divided by the total volume of the region of interest.	µm^2^/µm^3^
Trabecular thickness	Tb.Th	Mean thickness of trabeculae, calculated using direct 3D methods.	µm
Trabecular separation	Tb.Sp	Mean spacing between trabeculae, calculated using direct 3D methods.	µm
Trabecular bone pattern factor	Tb.Pf	Indicator of the spatial structure of bone. Relation between convex and concave elements. Tb.Pf < 0 when the trabecular bone is honeycomb-like and increases as the trabecular bone acquires a rod-like structure.	1/µm
Structure model index	SMI	Estimates the geometry of trabecular bone structure, with 0 for perfect plates, 3 for perfect rods, and 4 for perfect spheres	none
Total porosity (percent)	Po(tot)	The volume of pores within the trabecular structure as a percentage of the total VOI.	%
Connectivity	Conn.	It measures redundant connectivity, the degree to which parts of the object are multiply connected. It calculates the number of connections that can be severed before the structure can be divided into two separate pieces.	none

**Table 2 jcm-13-06679-t002:** Demographic data of the patients enrolled.

Test Group			Control Group		
Patients	Age	Sex	Patients	Age	Sex
Patient 1	48	female	Patient 8	69	female
Patient 2	60	male	Patient 9	42	male
Patient 3	43	female	Patient 10	48	female
Patient 4	73	female	Patient 11	46	male
Patient 5	53	male	Patient 12	33	male
Patient 6	55	female	Patient 13	69	male
Patient 7	52	female	Patient 14	44	male
			Patient 15	73	female

**Table 3 jcm-13-06679-t003:** Results of the micromorphometric analysis.

		Group Statistics		Independent Sample Test
				Sig.
		Mean	Std. Deviation	*p*
Percent bone volume (BVTV) *	test	22.088%	8.094%	0.000468 *
	control	12.075%	4.009%	
Bone surface/volume ratio (BS/TV)	test	0.018 µm^2^/µm^3^	0.007 µm^2^/µm^3^	0.457947
	control	0.016 µm^2^/µm^3^	0.004 µm^2^/µm^3^	
Trabecular thickness (Tb.Th)	test	267.349 µm	65.401 µm	0.838715
	control	272.282 µm	59.226 µm	
Trabecular separation (Tb.Sp) *	test	664.467 µm	218.084 µm	0.011763 *
	control	990.254 µm	366.158 µm	
Trabecular bone pattern factor (Tb.Pf)	test	0.0031/µm	0.0061/µm	0.199525
	control	0.0051/µm	0.0041/µm	
Structure model index (SMI) *	test	0.477	1.826	0.018648 *
	control	1.954	1.141	
Total porosity (percent) (Po.tot) *	test	77.912%	8.094%	0.000468 *
	control	87.925%	4.009%	
Connectivity (Conn.) *	test	2451.786	1524.472	0.005723 *
	control	1040.154	708.990	

* *p* < 0.05. Asterisk indicates that there is a significant difference between the values. Tb.Sp values of the two study groups were compared using the Mann–Whitney U test. Independent sample t-test was performed for the comparisons of all other parameters among study groups.

**Table 4 jcm-13-06679-t004:** Results of the histomorphometric analysis. Independent sample t-test was performed to compare BVP between study groups.

		Group Statistics		Independent Sample Test
				Sig.
		Mean	Std. Deviation	*p*
Bone Volume Percent (BVP)	test	47.64%	10.766%	0.262805
	control	42.70%	10.716%	

## Data Availability

Data are contained within the Appendix A.

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
