# Peer review of "Microarchitectural Study of the Augmented Bone Following a Modified Ridge Splitting Technique: Histological and Micro-Computed Tomography Analyses"

_jcm, 2024, doi:10.3390/jcm13226679_

Round 1

Reviewer 1 Report

Comments and Suggestions for Authors

Introduction:

·       Most of the references in the introduction part are before 2020 (need to add more recent references)

·       Literature of microarchitectural for other ridge augmentation techniques

Material and Methods:

study design

·       No demographic data for the patients 

·       No mention how many diabetic patients were in control and test group

·       The author didn’t mention the methods of patient assignment to control and test group (randomization method)

Tooth Extraction in control group:

·       Need to mention the cause of teeth extraction

Bone Biopsy samples harvesting:

·       There was no preoperative surgical stent to determine bone core biopsy from grafted or socket area

Discussion needs to be more comprehensive and give examples of studies who have done similar to your topic and compare them their results with yours

Author Response

We would like to thank the reviewer for their time and effort in reviewing the manuscript. The corrections suggested in the review report allow us to improve the manuscript so that its quality may reach the high standards of this esteemed Journal.

Comment 1: Introduction: Most of the references in the introduction part are before 2020 (need to add more recent references)

Response 1: We have updated the references. We inserted several references which were published after 2020.

[24] Báskay J, Pénzes D, Kontsek E, Pesti A, Kiss A, Guimarães Carvalho BK et al. Are Artificial Intelligence-Assisted Three-Dimensional Histological Reconstructions Reliable for the Assessment of Trabecular Microarchitecture? Journal of clinical medicine. 2024;13.

[32] Bonato RS, Fernandes GVO, Calasans-Maia MD, Mello A, Rossi AM, Carreira ACO et al. The Influence of rhBMP-7 Associated with Nanometric Hydroxyapatite Coatings Titanium Implant on the Osseointegration: A Pre-Clinical Study. Polymers. 2022;14.

[31] Alqahtani AR, Desai SR, Patel JR, Alqhtani NR, Alqahtani AS, Heboyan A et al. Investigating the impact of diameters and thread designs on the Biomechanics of short implants placed in D4 bone: a 3D finite element analysis. BMC oral health. 2023;23:686.

[25] Gasperini FM, Fernandes GVO, Mitri FF, Calasans-Maia MD, Mavropoulos E, Malta Rossi A et al. Histomorphometric evaluation, SEM, and synchrotron analysis of the biological response of biodegradable and ceramic hydroxyapatite-based grafts: from the synthesis to the bed application. Biomedical materials (Bristol, England). 2023;18.

[18] Tan J, Labrinidis A, Williams R, Mian M, Anderson PJ, Ranjitkar S. Micro-CT-Based Bone Microarchitecture Analysis of the Murine Skull. Methods in molecular biology (Clifton, NJ). 2022;2403:129-45.

[17] Fernandes GVO, Castro F, Pereira RM, Teixeira W, Gehrke S, Joly JC et al. Critical-size defects reconstruction with four different bone grafts associated with e-PTFE membrane: A histomorphometric experimental in vivo study. Clinical oral implants research. 2024;35:167-78.

[11] Shaker AE, Salem AS, El-Farag SA, Abdel-Rahman FH, El-Kenawy MH. Comparison of Khoury's Bone Shell Technique vs Titanium-reinforced Polytetrafluoroethylene Membrane for 3D-bone Augmentation in Atrophic Posterior Mandible: A Randomized Clinical Trial. The journal of contemporary dental practice. 2024;25:518-26.

[9] Ren Y, Fan L, Alkildani S, Liu L, Emmert S, Najman S et al. Barrier Membranes for Guided Bone Regeneration (GBR): A Focus on Recent Advances in Collagen Membranes. International journal of molecular sciences. 2022;23.

[10] Tumedei M, Mourão CF, D’Agostino S, Dolci M, Di Cosola M, Piattelli A et al. Histological and Histomorphometric Effectiveness of the Barrier Membranes for Jawbone Regeneration: An Overview of More Than 30 Years’ Experience of Research Results of the Italian Implant Retrieval Center (1988–2020). 2021;11:2438.

[13] Luo F, Mo Y, Jiang J, Wen J, Ji Y, Li L, Wan Q. Advancements in dental implantology: The alveolar ridge split technique for enhanced osseointegration. Clin Implant Dent Relat Res. 2024 Oct;26(5):1012-1031. doi: 10.1111/cid.13363. Epub 2024 Jul 29. PMID: 39075020.

[14] Scarano, A., Santos de Oliveira, P., Tagariello, G., Dipalma, G., Greco Lucchina, A., Mortellaro, C., ... & Lorusso, F. (2023). Rehabilitation of patients with thin ridges by conical expanders and immediate cone morse dental implant: a case report. European Review for Medical and Pharmacological Sciences, 27(3 Suppl), 141-146.

[15] Al Shawaheen, A., Khashaba, M., Elfaramawi, T., & Saleh, H. (2022). Evaluation of Bone Width Recovery After Using Modified Ridge Splitting Technique With and Without Beta-Tricalcium Phosphate Material (A randomized clinical trial). Egyptian Dental Journal, 68(4), 3141-3150.

[16] Issa, D. R., Elamrousy, W., & Gamal, A. Y. (2024). Alveolar ridge splitting and simvastatin loaded xenograft for guided bone regeneration and simultaneous implant placement: randomized controlled clinical trial. Clinical Oral Investigations, 28(1), 71

Comment 2: Introduction: Literature of microarchitectural for other ridge augmentation techniques

Response 2: We have added other ridge augmentation techniques in the introduction part. We have added this part to the introduction in row 60-61: “Guided bone regeneration and onlay block grafts have been applied successfully to restore the horizontally deficient alveolar ridge.”

Comment 3: Material and Methods: study design: No demographic data for the patients

Response 3: The demographic data was corrected in the Results section: “. Seven patients were included in the test group (two males five females, mean age 54,86 years ± 9,62 years, 14 bone core biopsy samples harvested), eight patients were allocated to the control group (five males, three females, mean age 53,25 years ± 15,06 years, 13 bone core biopsy samples harvested).” We have added a table (Table 2.) with all demographic data to the manuscript.

Comment 4: Material and Methods: study design: No mention how many diabetic patients were in control and test group

Response 4: None of the patients in either study groups had diabetes mellitus. We have added the following to the Results section: 

"None of the patients enrolled had diabetes mellitus in their medical history."

Comment 5: Material and Methods: study design: The author didn’t mention the methods of patient assignment to control and test group (randomization method)

Response 5: The determining factor of the randomization was the detected alveolar bone width. “Patients with insufficient bone width for dental implant placement at the premolar and molar region of the mandible were included in the test group” in row 103-104. “Patients with sufficient bone width for dental implant placement at the premolar and molar region of the mandible following 3-month-long spontaneous healing following tooth removal were included in the control group. Patients of the control group were included from a pool to match the age distribution of the patients included in the test group” in row 113-115.

Comment 6: Tooth Extraction in control group:  Need to mention the cause of teeth extraction

Response 6: We have added the cause of teeth extraction. In row 115-117 we have added: “Unrestorable teeth, teeth with hopeless prognosis due to periodontal disease, endodontic failures, and vertical root fracture were the indications of tooth removal”

Comment 7: Bone Biopsy samples harvesting: There was no preoperative surgical stent to determine bone core biopsy from grafted or socket area

Response 7: In this study, dental implants were placed free-hand. However, prosthetically driven implant placement was performed by an experienced surgeon. Computer Assisted Implant Placement could have facilitated the harvesting of bone core biopsy samples and prosthetically driven implant placement. However, the extent of the bone defects in both the test and control groups guaranteed that the augmented bone and extraction sockets were sampled.

Comment 8: Discussion needs to be more comprehensive and give examples of studies who have done similar to your topic and compare them their results with yours

Response 8: We have carefully reviewed the literature and found that the most prevalent approach to ridge splitting is grafting the bone defects and performing dental implant placement simultaneously. In this case, authors do not harvest biopsy samples following bony healing due to ethical concerns as this would include a re-entry after the bony healing of the implants. This would serve no benefit for the patients, however, it would expose them to another round of surgery. In the studies where bone blocks are used as graft material for ridge splitting, histological evaluation was performed only by Scipioni et al. They examined the micromorphology of the augmented areas following a graftless ridge splitting procedure using histological methods. After the 16-month healing period a mature, regenerated bone was observed. To the best of our knowledge, the present study is the first to evaluate the quality and microarchitecture of augmented bone following staged alveolar ridge splitting using autologous bone blocks, based on histomorphometric and micro-CT analysis. 

"Scipioni et al. 41 observed new bone formation following a graftless edentulous ridge expansion based on the results of histologic analysis of bone core biopsy samples. To the best of our knowledge, the present study is the first to evaluate the quality and microarchitecture of augmented bone following staged alveolar ridge splitting using autologous bone blocks, based on histomorphometric and micro-CT analysis."

The authors would like to thank the reviewer for pointing out the shortcomings of the manuscript. The authors hope that the changes made in the manuscript are sufficient and the reviewer recommends our paper for publishing in the Journal.

Reviewer 2 Report

Comments and Suggestions for Authors

Dear authors, I want to thank you for having submitted your article to our journal; i believe some revisions are needed before the article can be accepted.

1. No images are available regarding the histology; at least a couple of slides are needed.

2. An image of the donor site showing how the bone block was harvested and how the bone block was adapted to the recipient site are needed.

3. More data are needed on the patients in the control group, as the conditions of the bone ridge before the extraction might have had an effect on the bone remodeling process.

4. All acronyms must be explained in the tables, as tables should be comprehensible even without reading the rest of the article.

Author Response

We would like to thank the reviewer for their time and effort in reviewing the manuscript. The corrections suggested in the review report allows us to improve the manuscript so that its quality may reach the high standards of this esteemed Journal.

Comment 1: 1. No images are available regarding the histology; at least a couple of slides are needed.

 Response 1: We have added Figure 7. A typical histological appearance of a bone core biopsy is illustrated in several magnifications.

Comment 2: An image of the donor site showing how the bone block was harvested and how the bone block was adapted to the recipient site are needed.

Response 2: We have added a figure ( Figure 1) where the procedure of bone block harvesting and fixation is showed.

Comment 3: More data are needed on the patients in the control group, as the conditions of the bone ridge before the extraction might have had an effect on the bone remodeling process.

Response 3: We have assigned the cause of teeth extraction in row 115-117: “Unrestorable teeth, teeth with hopeless prognosis due to periodontal disease, endodontic failures, and vertical root fracture were the indications of tooth removal.” We have not monitored the conditions of the bone before teeth extractions. Only 2-D X-ray recordings were used with clinical examinations to decide for extraction.

Comment 4: All acronyms must be explained in the tables, as tables should be comprehensible even without reading the rest of the article.

Response 4: All acronyms are explained in the tables.

The authors would like to thank the reviewer for pointing out the shortcomings of the manuscript. The authors hope that the changes made in the manuscript are sufficient and the reviewer recommends our paper for publishing in the Journal.

Reviewer 3 Report

Comments and Suggestions for Authors

Dear authors,
I evaluated the study titled “Microarchitectural Study of the Augmented Bone Following a Modified Ridge Splitting Technique, Histological, and Micro-Computed Tomography Analysis”. This study involved 15 authors, which I considered a high number.

The goal of the study was “to examine the microarchitecture of the augmented bone following a modified alveolar ridge splitting procedure and compare it to that of native bone.”

The study is interesting and is presenting a modified ridge splitting technique (it can be a limitation for the inclusion of patients added to the eligibility criteria used). The authors include it in the limitation of the study.

Some comments:
- Where is the sample size calculation? I’m asking it because the final number of patients observed is low and the same of the number of authors. My suggestion is to present a sample size calculation or

- For reference 12 (histomorphometric evaluation), please change it for this ref. (better to the goal of the study):
(new ref) Critical-size defects reconstruction with four different bone grafts associated with e-PTFE membrane: A histomorphometric experimental in vivo study. Clin Oral Impl Res. 2024; 35(2):167-178. doi: 10.1111/clr.14210

- with the ref. 13, include this article below to show that the method has been used for long time
(new ref.) Influence of estrogen deficiency and tibolone therapy on trabecular and cortical bone evaluated by computed radiography system in rats. Acta Cir Br, 2012,27(3), 217-222. doi: 10.1590/s0102-86502012000300003

- Fig. 1 must be moved to the Results section

- The description of the surgical technique was clear and well-performed.

- Results: please improve the description of the histomorphometric analysis.
Please move the text at the Discussion section (lines 301-313) to Results.

- Why the authors did not had another control group applying the technique without modification?

- Line 314 (Discussion): with ref. 19, include the reference below:
(new ref) Histomorphometric evaluation, SEM, and synchrotron analysis of the biological response of biodegradable and ceramic hydroxyapatite-based grafts: from the synthesis to the bed application. Biomed Mater. 18 (2023), 065023. https://doi.org/10.1088/1748-605X/ad0397

- line 332: with the refs. used “recipient site is a risk factor of early implant failure [9-11]”, include the ref. below.
(new ref.) Investigating the Impact of Diameters and Thread Designs on the Biomechanics of Short Implants Placed in D4 Bone: A 3D Finite Element Analysis. BMC Oral Health, 2023;23:686. https://doi.org/10.1186/s12903-023-03370-8

- Lines 335-340: there is no reference. I suggest to include the ref. below showing the evolution for the osseointegration process.
(new ref.) The Influence of rhBMP-7 Associated with Nanometric Hydroxyapatite Coatings Titanium Implant on the Osseointegration: A Pre-Clinical Study. Polymers, 2022, 14(19), 4030; https://doi.org/10.3390/polym14194030

- Only four references used in the article (13%) were published in the last five years. Update it (I sent some articles above)

Author Response

We would like to thank the reviewer for their time and effort in reviewing the manuscript.

Comment 1: I evaluated the study titled “Microarchitectural Study of the Augmented Bone Following a Modified Ridge Splitting Technique, Histological, and Micro-Computed Tomography Analysis”. This study involved 15 authors, which I considered a high number.

 Response 1: The number of the authors is explained by the complexity of this manuscript. It consists not only of surgical methods but also microCT, histological examination, and the use of Artificial Intelligence for histomorphometry. Nevertheless, according to the guidelines, all professionals listed qualify for authorship on this manuscript.

Comment 2: The goal of the study was “to examine the microarchitecture of the augmented bone following a modified alveolar ridge splitting procedure and compare it to that of native bone.”
The study is interesting and is presenting a modified ridge splitting technique (it can be a limitation for the inclusion of patients added to the eligibility criteria used). The authors include it in the limitation of the study. Where is the sample size calculation? I’m asking it because the final number of patients observed is low and the same of the number of authors. My suggestion is to present a sample size calculation

Response 2: We have included a sample size calculation in row 151-156. : Sample size calculation was performed using the G*Power 3.1 software (v.3.1.9.3, 2017, Institut für Experimentelle Psychologie, Heinrich-Heine-Universität, Düsseldorf, Germany); 0.1 (10%) difference in BV/TV was considered as clinically significant. If power was set at 0.8 and α at 0.05, a sample size of 9 cases (bone core biopsy samples) per study group was calculated as sufficient.

Comment 3: For reference 12 (histomorphometric evaluation), please change it for this ref. (better to the goal of the study): (new ref) Critical-size defects reconstruction with four different bone grafts associated with e-PTFE membrane: A histomorphometric experimental in vivo study. Clin Oral Impl Res. 2024; 35(2):167-178. doi: 10.1111/clr.14210.

Response 3: Thank you. We have updated the references.

Comment 4: Fig. 1 must be moved to the Results section

Response 4: This figure was moved to the Results section.

Comment 5: The description of the surgical technique was clear and well-performed.

Response 5: Thank you!

Comment 6: Results: please improve the description of the histomorphometric analysis .

Response 6: We have revised the description of the histomorphometric analysis: “An in-house Artificial Intelligence (AI) application was used to classify tissues (trabecular bone and bone marrow) in the histological sections 28. Three representative sections from each bone biopsy sample were evaluated, and the areas of trabecular bone and bone marrow were measured using the AI model. An expert (E.K.) supervised the automatic segmentation and made sure that the tissues were identified correctly. Bone Volume Percent (BVP) was determined as the primary outcome measure of the histomorphometric analysis which is the ratio (%) of the area of bone within the total area of the histological section. BVP was calculated as the mean of the measurements performed on the three representative histological sections for each bone biopsy specimen.”

Comment 7: Please move the text at the Discussion section (lines 301-313) to Results.

Response 7: The part of the text was moved to the Result section.

Comment 8: Why the authors did not had another control group applying the technique without modification?

Response 8: The conventional approach to ridge splitting is to place dental implants simultaneously using graft materials as fillers in the bone defect created during the procedure. The modified approach described in this study is staged and allows for the harvesting of a bone core biopsy sample from the dental implant recipient site without additional surgical burden to the patient. In the case of the conventional ridge splitting, biopsy samples could only be harvested from a healed site without implant placement, which is a further burden to the patient with no benefits in their treatment. This would be ethically questionable. Therefore, we have opted against including a control group with the conventional ridge-splitting method.

Comment  9: Line 314 (Discussion): with ref. 19, include the reference below: (new ref) Histomorphometric evaluation, SEM, and synchrotron analysis of the biological response of biodegradable and ceramic hydroxyapatite-based grafts: from the synthesis to the bed application. Biomed Mater. 18 (2023), 065023. https://doi.org/10.1088/1748-605X/ad0397

Response 9: We have added this reference to the manuscript.

Comment 10: line 332: with the refs. used “recipient site is a risk factor of early implant failure [9-11]”, include the ref. below. (new ref.) Investigating the Impact of Diameters and Thread Designs on the Biomechanics of Short Implants Placed in D4 Bone: A 3D Finite Element Analysis. BMC Oral Health, 2023;23:686. https://doi.org/10.1186/s12903-023-03370-8

Response 10: We have added this reference to the manuscript.

Comment  11: Lines 335-340: there is no reference. I suggest to include the ref. below showing the evolution for the osseointegration process. (new ref.) The Influence of rhBMP-7 Associated with Nanometric Hydroxyapatite Coatings Titanium Implant on the Osseointegration: A Pre-Clinical Study. Polymers, 2022, 14(19), 4030; https://doi.org/10.3390/polym14194030

Response 11: We have added this refence to the manuscript.

Comment 12: Only four references used in the article (13%) were published in the last five years. Update it (I sent some articles above)

Response 12: We have updated the references. We have added more references published in the last five years to the  .

[24] Báskay J, Pénzes D, Kontsek E, Pesti A, Kiss A, Guimarães Carvalho BK et al. Are Artificial Intelligence-Assisted Three-Dimensional Histological Reconstructions Reliable for the Assessment of Trabecular Microarchitecture? Journal of clinical medicine. 2024;13.

[32] Bonato RS, Fernandes GVO, Calasans-Maia MD, Mello A, Rossi AM, Carreira ACO et al. The Influence of rhBMP-7 Associated with Nanometric Hydroxyapatite Coatings Titanium Implant on the Osseointegration: A Pre-Clinical Study. Polymers. 2022;14.

[31] Alqahtani AR, Desai SR, Patel JR, Alqhtani NR, Alqahtani AS, Heboyan A et al. Investigating the impact of diameters and thread designs on the Biomechanics of short implants placed in D4 bone: a 3D finite element analysis. BMC oral health. 2023;23:686.

[25] Gasperini FM, Fernandes GVO, Mitri FF, Calasans-Maia MD, Mavropoulos E, Malta Rossi A et al. Histomorphometric evaluation, SEM, and synchrotron analysis of the biological response of biodegradable and ceramic hydroxyapatite-based grafts: from the synthesis to the bed application. Biomedical materials (Bristol, England). 2023;18.

[18] Tan J, Labrinidis A, Williams R, Mian M, Anderson PJ, Ranjitkar S. Micro-CT-Based Bone Microarchitecture Analysis of the Murine Skull. Methods in molecular biology (Clifton, NJ). 2022;2403:129-45.

[17] Fernandes GVO, Castro F, Pereira RM, Teixeira W, Gehrke S, Joly JC et al. Critical-size defects reconstruction with four different bone grafts associated with e-PTFE membrane: A histomorphometric experimental in vivo study. Clinical oral implants research. 2024;35:167-78.

[11] Shaker AE, Salem AS, El-Farag SA, Abdel-Rahman FH, El-Kenawy MH. Comparison of Khoury's Bone Shell Technique vs Titanium-reinforced Polytetrafluoroethylene Membrane for 3D-bone Augmentation in Atrophic Posterior Mandible: A Randomized Clinical Trial. The journal of contemporary dental practice. 2024;25:518-26.

[9] Ren Y, Fan L, Alkildani S, Liu L, Emmert S, Najman S et al. Barrier Membranes for Guided Bone Regeneration (GBR): A Focus on Recent Advances in Collagen Membranes. International journal of molecular sciences. 2022;23.

[10] Tumedei M, Mourão CF, D’Agostino S, Dolci M, Di Cosola M, Piattelli A et al. Histological and Histomorphometric Effectiveness of the Barrier Membranes for Jawbone Regeneration: An Overview of More Than 30 Years’ Experience of Research Results of the Italian Implant Retrieval Center (1988–2020). 2021;11:2438.

[13] Luo F, Mo Y, Jiang J, Wen J, Ji Y, Li L, Wan Q. Advancements in dental implantology: The alveolar ridge split technique for enhanced osseointegration. Clin Implant Dent Relat Res. 2024 Oct;26(5):1012-1031. doi: 10.1111/cid.13363. Epub 2024 Jul 29. PMID: 39075020.

[14] Scarano, A., Santos de Oliveira, P., Tagariello, G., Dipalma, G., Greco Lucchina, A., Mortellaro, C., ... & Lorusso, F. (2023). Rehabilitation of patients with thin ridges by conical expanders and immediate cone morse dental implant: a case report. European Review for Medical and Pharmacological Sciences, 27(3 Suppl), 141-146.

[15] Al Shawaheen, A., Khashaba, M., Elfaramawi, T., & Saleh, H. (2022). Evaluation of Bone Width Recovery After Using Modified Ridge Splitting Technique With and Without Beta-Tricalcium Phosphate Material (A randomized clinical trial). Egyptian Dental Journal, 68(4), 3141-3150.

[16] Issa, D. R., Elamrousy, W., & Gamal, A. Y. (2024). Alveolar ridge splitting and simvastatin loaded xenograft for guided bone regeneration and simultaneous implant placement: randomized controlled clinical trial. Clinical Oral Investigations, 28(1), 71

The authors would like to thank the reviewer for pointing out the shortcomings of the manuscript. The authors hope that the changes made in the manuscript are sufficient and the reviewer recommends our paper for publishing in the Journal.

Reviewer 4 Report

Comments and Suggestions for Authors

Dear Authors,

I have read your manuscript entitled "Microarchitectural study of the augmented bone following a modified ridge splitting technique, histological, and micro-computed tomography analysis ".

The study presented has the limitation of a small sample size, as stated by the authors. The technique proposed is interesting and well exposed.

Was the surgeon the same in all the cases reported? If yes, please declare it in materials and methods and add his/her initials (P.D.) also in this section. If not please add it to the limitations of the study.

Did the authors made a correlation between the bone parametres investigated and sex/age of the sample? If no, please add it to the limitations.

Did the authors used a pain score to evaluate the post-operative period of time? If no, please add it to the limitations.

Please delete the Acknowledge section if you do not need it.

I think the introduction will benefit of a couple of additional topic. One regards the possibility of the use of other solution for bone regeneration, a complete overview about barrier membranes is provided by the following reference: https://www.mdpi.com/2076-3417/11/5/2438.

Finally, a brief mention should be done about the role of the post-estractive implant placement, its clinical relevance is provided by the following reference: https://pubmed.ncbi.nlm.nih.gov/11759868/.

Author Response

We would like to thank the reviewer for their time and effort to review the manuscript.

Comment 1: I have read your manuscript entitled "Microarchitectural study of the augmented bone following a modified ridge splitting technique, histological, and micro-computed tomography analysis ". The study presented has the limitation of a small sample size, as stated by the authors. The technique proposed is interesting and well exposed.

Response 1: Thank you.

Comment 2: Was the surgeon the same in all the cases reported? If yes, please declare it in materials and methods and add his/her initials (P.D.) also in this section. If not please add it to the limitations of the study.

Response 2: We have declared it in row 150-151: “All surgical interventions were performed by the same surgeon experienced in dental implant placement and bone management (D.P.).”

Comment 3: Did the authors made a correlation between the bone parametres investigated and sex/age of the sample? If no, please add it to the limitations.

Response 3: We have this limitation to the article in row 371-373:  “Another limitation of this study was that parameters of trabecular bone microarchitecture were not correlated with the sex and age of the patients enrolled, because of the small sample size.”

Comment 4: Did the authors used a pain score to evaluate the post-operative period of time? If no, please add it to the limitations.

Response 4: We have this limitation to the article in row 374-375: “A further limitation was that patient-reported postoperative pain was not recorded following the surgical interventions.”

Comment 5: Please delete the Acknowledge section if you do not need it.

Response 5:  The Acknowledge section was deleted.

Comment 6: I think the introduction will benefit of a couple of additional topic. One regards the possibility of the use of other solution for bone regeneration, a complete overview about barrier membranes is provided by the following reference: https://www.mdpi.com/2076-3417/11/5/2438.

Response 6: We have explained other augmentation strategies in the Introduction part in row 60-61: “Guided bone regeneration and onlay block grafts have been applied successfully to restore the horizontally deficient alveolar ridge.” The above mentioned reference was applied.

Comment 7: Finally, a brief mention should be done about the role of the post-estractive implant placement, its clinical relevance is provided by the following reference: https://pubmed.ncbi.nlm.nih.gov/11759868/.

Response 7: We have mentioned the early implant placement in row 53-55: “Following tooth removal, immediate implant placement may preserve the alveolar anatomy. “

The authors would like to thank the reviewer for pointing out the shortcomings of the manuscript. The authors hope that the changes made in the manuscript are sufficient and the reviewer recommends our paper for publishing in the Journal.

Round 2

Reviewer 3 Report

Comments and Suggestions for Authors

Dear authors,

Thank you so much for all your polite and precise responses.

I appreciate the revision provided.

Congratulations on the study.